# Morphological and Spectroscopic Study of an Apatite Layer Induced by Fast-Set Versus Regular-Set EndoSequence Root Repair Materials

**DOI:** 10.3390/ma12223678

**Published:** 2019-11-08

**Authors:** Sawsan T. Abu Zeid, Ruaa A. Alamoudi, Ensanya A. Abou Neel, Abeer A. Mokeem Saleh

**Affiliations:** 1Endodontic Department, Faculty of Dentistry, King Abdulaziz University, Jeddah 21589, Saudi Arabia; ralamoudi1@kau.edu.sa (R.A.A.); aasaleh@kau.edu.sa (A.A.M.S.); 2Endodontic Department, Faculty of Dentistry, Cairo University, Cairo 12613, Egypt; 3Division of Biomaterials, Restorative Dentistry Department, Faculty of Dentistry, King Abdulaziz University, Jeddah 21589, Saudi Arabia; eabouneel@kau.edu.sa; 4Biomaterials Department, Faculty of Dentistry, Tanta University, Tanta 31512, Egypt; 5UCL Eastman Dental Institute, Biomaterials and Tissue Engineering Division, 256 Gray’s Inn Road, London WC1X 8LD, UK

**Keywords:** apatite formation, bioactivity, Ca/P and CO_3_/PO_4_ ratios, root repair, spectroscopy analysis

## Abstract

This study aimed to evaluate the morphology and chemistry of an apatite layer induced by fast-set versus regular-set EndoSequence root repair materials using spectroscopic analysis. Holes of a 4 mm diameter were created in the root canal dentin, which were filled with the test material. Fetal calf serum was used as the incubation medium, and the samples incubated in deionized water were used as controls. The material-surface and material-dentin interfaces were analyzed after 28 days using Raman and infrared spectroscopy, scanning electron microscopy/energy dispersive X-ray, and X-ray diffraction. After incubation in fetal calf serum, both materials formed a uniform layer of calcium phosphate precipitate on their surfaces, with the dentinal interface. This precipitated layer was a combination of hydroxyapatite and calcite or aragonite, and had a high mineral maturity with the regular-set paste. However, its crystallinity index was high with the fast-set putty. Typically, both consistencies (putty and paste) of root repair material have an apatite formation ability when they are incubated in fetal calf serum. This property could be beneficial in improving their sealing ability for root canal dentin.

## 1. Introduction

Root perforation is an undesired iatrogenic endodontic mishap. It may occur on a lateral root surface or in a furcal area. It provides direct communication between pulp space and periodontal tissue, and leads to a loss of root integrity and endodontic failure [1]. Since the early 1990s, mineral trioxide aggregates (MTA) have been a popular material in root repair [2]. The success of perforation repair is dependent on the material’s ability to seal the perforation defect [3].

Different root repair materials have been introduced into the market to overcome the drawbacks of traditional mineral trioxide aggregates, such as the long setting time and poor handling. EndoSequence root repair materials (ERRM) are mainly composed of calcium silicate and calcium phosphate [4]. As an alternative to MTA, calcium silicate-based materials have been considered bioactive, which is the material’s ability to form an apatite-like precipitate on its surface when brought into contact with tissue fluids. This precipitation occurs as a result of the interaction of ions released from the material with the surrounding tissue fluid [5]. Apatite precipitation can improve the sealing ability of these materials when occurring at a material-dentin interface [5,6].

Different formulas of ERRM have been launched to improve their application. ERRM are available with different consistencies: fast-set putty with a heavy consistency and regular-set paste with a light consistency. These different consistencies exhibit variable physical behavior as they have various particle sizes [7]. Fast-set putty exhibits an adhesive failure mode and greater displacement resistance than regular-set paste [7]. Therefore, this difference may affect their chemical behavior and morphology when immersed in simulated tissue fluid.

There are few studies that have evaluated the surface of EndoSequence putty when immersed in different solutions. They have assessed the surface morphology of the material and apatite precipitation by scanning electron microscopy [8,9]. The surface topography of putty varied, according to the fluid in which the material was incubated. The unset putty material exhibited a smooth surface topography. After setting, the putty showed a smooth topography, with clusters of elongated crystals when incubated in water. Globular deposits, however, were found when set putty was incubated in Hank’s Balanced Salt Solution [8]. Another study evaluated the surface nucleation of TotalFill putty and paste formula. Both formulas showed different chemical behaviors [10].

No study has evaluated the material-dentin interface, which is a significant parameter for detecting the sealing ability of any material.

This study evaluates the morphology and chemistry of an apatite layer induced by fast-set versus regular-set EndoSequence root repair materials after incubation in simulated tissue fluid (e.g., fetal calf serum) [11]. The material, material-dentin, and dentin surfaces were evaluated using scanning electron microscopy/energy dispersive X-ray (SEM/EDX), Raman Spectroscopy, Fourier transform infrared spectroscopy (FTIR), and X-ray diffraction (XRD) [6,12,13].

The proposed null hypothesis states that “there is no difference between the fast-set putty and regular-set paste root repair materials in terms of apatite formation”.

## 2. Materials and Methods

EndoSequence Brasseler root repair materialBC RRM fast-set putty and regular-set paste root repair materials (Brasseler, Boulevard, Savannah, GA, USA) were used for this study.

### 2.1. Specimen Preparation

After getting ethical approval (#063-04-18), human, freshly extracted, single-rooted premolars were used for this study. The longitudinal root sections (n = 12) were randomly divided into two groups (n = 6 each) [10]. Two holes of a 4 mm diameter, representing lateral root perforation, were prepared in each root section and filled with EndoSequence fast-set putty or regular-set paste. They were then wrapped with gauze moistened with deionized water and incubated at 37 °C/100% relative humidity for three days, until the complete setting of the material. They were then subdivided into two groups (n = 3 each): the experimental and control group. The samples of the experimental group were incubated in fetal calf serum (FCS, Gibco, Life Technology, Carlsbad, CA,, USA) for 28 days, with the FCS replaced at 3 day intervals, while the samples of the control group were incubated in deionized water. At the end of the experiment, the samples were washed with deionized water and left to dry for 48 h in an incubator. They were then analyzed with Raman Spectroscopy, Fourier transform infrared spectroscopy (FTIR), and a scanning electron microscope/energy dispersive X-ray (SEM/EDX).

Discs (n = 8) of a 10 mm diameter and 3 mm height from EndoSequence fast-set putty and regular-set paste root repair materials were prepared and left to set as described above. These discs were divided into two groups (n = 4 each); two were left in FCS for 28 days and the other two were used as controls (incubated in deionized water). These discs were analyzed with X-ray diffraction (XRD).

### 2.2. Raman Spectroscopy

The spectra of the materials and the material-dentin interface were obtained using Micro-Raman Spectroscopy (Senterra, Bruker, Berlin, Germany) associated with a laser driver (Nd-YAG dropped AIY Jernet), CCD detector (Bruker, Germany), and microscope with a confocal pinhole. Three spectra from different locations were obtained for each sample. The ratio of 965Ap/916CaWO_4_ cm^−1^ band intensity as an indication of the thickness of the apatite layer at the material-surface and the material-dentin interface was calculated [14,15,16]. The mineral maturity was also considered, and it was calculated as the area ratio of the apatitic phosphate band detected at 1030 cm^−1^/non-apatitic phosphate band detected at 1110 cm^−1^ [14,15,16].

### 2.3. Fourier Transform Infrared Spectroscopy

The spectra of the material-surface and the material-dentin interface were obtained using Fourier Transform Infrared Spectroscopy (FTIR-6100, Jasco, Tokyo, Japan). The spectra were recorded in the 4000–400 cm^−1^ range and at a 1 cm^−1^ resolution. Three spectra from different locations were obtained for each sample. The carbonate/phosphate (CO_3_/PO_4_) ratio, which is an indication of carbon substitution for phosphate, was calculated from the integrated area under carbonate (v3 CO_3_) and phosphate (ν1 ν3 PO_4_) bands at 830–890 and 900–1200 cm^−1^, respectively [13,17]. The crystallinity index, an indication of the crystal perfection, was calculated using the asymmetric stretching vibration of the phosphate bands at 604 and 560 cm^−1^ [14,18].

### 2.4. Scanning Electron Microscope/Energy Dispersed X-ray Analysis (SEM/EDX)

A scanning electron microscope/energy dispersed X-ray analysis (Quanta 250 Field Emission Gun attached, FEI Company, Eindhoven, The Netherlands, UK) was used to study the morphology and elemental composition of each sample. Three EDX readings were obtained from three different areas across the sample. From the EDX, the calcium/phosphate (Ca/P) ratio was calculated from the atomic percentage.

### 2.5. X-ray Diffraction (XRD)

The discs of each material were milled into a fine powder and analyzed by XRD (Empyrean, Analytical 2010, Eindhoven, The Netherlands, UK).

### 2.6. Statistical Analyses

Statistical analysis was performed using a Student t-test. SPSS software (Version 16, Munich, Germany) was used at the significance level of 5%.

## 3. Results

### 3.1. Raman Spectroscopy

The Raman spectra of EndoSequence fast-set putty and regular-set paste, kept in deionized water for 28 days, revealed nearly identical bands, which varied in their intensities (Figure 1a). The bands include ν3 carbonate near 1400 cm^−1^ [19]; calcite or aragonite at 1088 cm^−1^ [12,20,21,22]; sulfate (ν1 SO_4_^2−^ and ν2 SO_4_^2−^) at 1005 cm^−1^ and 421 cm^−1^, respectively [22,23]; phosphate (ν1 PO_4_^3−^) at 903–990 cm^−1^ [24]; crystalline silicate (SiO_4_^4−^) at 857 cm^−1^ [23]; and ν3 SiO_4_^4−^ at 535 and 520 cm^−1^ [22]. The intensities of the ν1 PO_4_^3−^ band at 927 cm^−1^ and ν4 SO_4_^2−^ at 633 and 614 cm^−1^ were higher with the EndoSequence fast-set putty than with the regular-set paste. Aluminum constituents at 714 cm^−1^ [25]; tantalum oxide around 660, 633, 613, 331, 252, and 197 cm^−1^ [26,27]; zirconia at 475 and 182 cm^−1^ [16]; organic bands of amide I (C=O stretch) at 1666 cm^−1^; and amide III (N-H) at 1230–1289 cm^−1^ [28] were also detected in the spectra of both materials.

After incubation in FCS for 28 days, apatitic phosphate bands were detected at the following values: ν3 PO_4_^3−^ at 1050, 1040, and 1030 cm^−1^; ν1 PO_4_^3− ^shift around 965 cm^−1^ [16,29]; ν4 PO_4_^3− ^at 588 cm^−1^; and ν2 PO_4_^3−^ around 430 cm^−1^ [23,28] (Figure 1b–g). The non-apatitic phosphate band was seen around 1110 cm^−1^ [15]. Additionally, the spectra of the interface revealed dentin bands at 961, 599, 520, and 430 cm^−1^ (Figure 1f,g).

The ratio of the 965Ap/916CaWO_4_ cm^−1^ band intensity at the material-surface and the material-dentin interface of EndoSequence fast-set putty was 1.19 ± 0.13 and 1.31 ± 0.11, respectively. This was compared to the respective values of 3.26 ± 0.35 and 5.17 ± 0.57 obtained at the surface and dentin interface of the regular-set paste.

After incubation in FCS, the mineral maturity at the putty surface and interface (1.53 ± 0.28 and 4.05 ± 0.79, respectively) was significantly (*p* = 0.004) lower than that of the paste (12.47 ± 5.57 and 14.66 ± 5.03, respectively).

### 3.2. Fourier Transform Infrared Spectroscopy

The infrared spectra of EndoSequence fast-set putty and regular-set paste, kept in deionized water for 28 days, showed nearly identical compositions, but with little variations in band intensities (Figure 2a). The spectra were characterized by the presence of intense carbonate bands (calcite or aragonite) at 1412 and 873 cm^−1^ [6,12,20]; however, these bands were more intense in the EndoSequence regular-set paste than in fast-set putty. The ν3 PO_4_^3−^ bands at 1060 and 1027 cm^−1^ [14,15,30] were also broad in the regular-set paste, and weak in the fast-set putty. The ν3 of polymerized silicate hydrate at 990 cm^−1^ [31], alite at 935 and 930 cm^−1^ [21,25,31], aluminate phase at 745 and 655 cm^−1^ [25,32], and calcium silicate hydrate at 506 and 445 cm^−1^ [6,21] was also detected for both materials. After incubation in FCS for 28 days, PO_4_^3−^ bands of ν1 around 960 cm^−1^, characteristic apatite twin bands of ν4 PO_4_^3−^ at 602 and 560 cm^−1^, and ν3 mode at 1027 cm^−1^ [23,28] were also detected (Figure 2b).

The EndoSequence fast-set putty showed a statistically insignificantly (*p* = 0.13) higher CO_3_/PO_4_ ratio (0.38 ± 0.05) than the regular-set paste (0.32 ± 0.07). However, after incubation in FCS, the CO_3_/PO_4_ for both materials was significantly decreased (0.22 ± 0.03 for putty and 0.27 ± 0.01 for paste), with no significant difference between them (*p* = 0.215).

The crystallinity index of the apatite precipitate seen on the surface of each material was 6.6 ± 1.3 and 4.35 ± 0.6 for EndoSequence fast-set putty and EndoSequence regular-set paste, respectively. There was no significant difference (*p* = 0.139) between them regarding the crystallinity index.

### 3.3. Scanning Electron Microscopy/Energy Dispersed X-ray Analysis

Figure 3 shows the SEM/EDX of EndoSequence fast-set putty and regular-set paste samples, kept in deionised water for 28 days. The surface of the EndoSequence fast-set putty had small, irregular particles (horizontal white arrows in Figure 3a,b), interrupted by small, rounded, globular crystals (horizontal black arrows in Figure 3b,c), and a few needle-like particles (vertical black arrows in Figure 3b,c). A few dispersed radio-opacifier dots (vertical white arrows in Figure 3a) were also seen. At the EndoSequence putty-dentin interface, a gap was clearly identified (Figure 3d). EDX analysis revealed the presence of calcium (Ca), oxygen (O), carbon (C), phosphorus (P), zirconium (Zr), tantalum (Ta), silicon (Si), and sodium (Na), with traces of fluoride (F), magnesium (Mg), aluminum (Al), and sulfur (S). The Ca/P atomic ratios were 6.58 and 7.12 for the material-surface and the material-dentin interface, respectively (Figure 3e,f). The surface of EndoSequence regular-set paste revealed the presence of small-sized particles and aggregates of medium-sized grains fused with the hydrogel film. Radio-opacifier dots, as well as a few scattered rod-like bundles, were also detected (Figure 3g–i). The paste-dentin interface was sealed with the material itself (Figure 3j). EDX analysis revealed a similar composition to EndoSequence fast-set putty, but with less Ca and more P, Zr, and Ta. The Ca/P atomic ratios were 2.32 and 2.83 for the material-surface and material-dentin interface, respectively (Figure 3k,l).

Figure 4 shows the SEM/EDX of EndoSequence fast-set putty and regular-set paste samples, kept in FCS for 28 days. A homogenous layer of flower-like, medium-sized crystals (vertical white arrow in Figure 4a,b) and small-sized crystals (horizontal white arrow in Figure 4c,d) covered the entire surface of the fast-set putty and regular-set paste, respectively. Both material-dentine interfaces were filled with spherulites and acicular crystals (Figure 4e,f). However, the respective Ca/P atomic ratios were 3.44 and 3.01 for material surfaces and 3.86 and 1.98 for the material-dentin interface of both fast-set putty and regular-set paste, respectively (Figure 4g–j).

### 3.4. X-ray Diffraction Analysis

The phase analysis revealed that the EndoSequence fast-set putty contained calcium carbonate [CaCO_3_, card No: 01-070-0095], sodium silicate hydroxide hydrate [Revdite, Na_16_(Si_4_O_6_(OH)_5_)_2_(Si_8_O_15_(OH)_6_)(OH)_10_(H_2_O)_28_, card No: 01-077-0852], magnesium aluminum silicate [pyrope, Mg_3_Al_2_(SiO_4_)_3_, card No: 01-083-1711], and calcium hydrogen phosphate hydrate [Ca(H_2_PO_4_)_2_H_2_O, card No: 01-075-1521]. Conversely, the EndoSequence regular-set paste contained calcium carbonate [CaCO_3_, card No: 00-003-0893], calcium phosphate [α-Ca_3_(PO_4_)_2_, card No: 00-029-0359], calcium silicate sulfate [Ternesite, Ca_5_(SiO_4_)_2_(SO_4_), card No: 01-088-0812], and sodium aluminum silicate [Na(AlSi_3_O_8_), card No: 01-071-6220]. Both materials contained tantalum oxide [Tantite O11, Ta_2_O_5_, card No: 01-071-0639] and zirconium oxide [Baddeleyite, ZrO_2_, card No: 01-070-2491] (Figure 5a,b). After incubation of the specimen in FCS, additional phases of calcium phosphate hydroxide [hydroxyapatite, Ca_5_(PO_4_)_3_(OH); card No: 01-073-8421] and calcite [CaCO_3_; card No: 01-075-4353] were observed in EndoSequence fast-set putty, while aragonite [CaCO_3_; card No: 00-003-0893] was detected in the regular-set paste (Figure 5c,d).

## 4. Discussion

This study aimed to evaluate the morphology and chemistry of an apatite layer induced by two different formulations of EndoSequence root repair material: the EndoSequence fast-set putty and regular-set paste. Apatite precipitation was detected over both materials after incubation in FCS for 28 days. Shokouhinejad et al., 2012 detected apatite precipitation on ERRM putty one week after immersion in phosphate buffer solution [9].

Raman and FTIR spectroscopy revealed that both formulations contain calcium carbonate (calcite or aragonite), silicate, phosphate, aluminum, sulfur, zirconium, and tantalum bands. However, the EndoSequence regular-set paste showed a more intense calcite band at 1412 cm^−1^ and a broader ν3 PO_4_^3−^ band at 1060 cm^−1^ compared to fast-set putty, which had a weak ν3 PO_4_^3−^ band at 1027 cm^−1^. This finding was also confirmed by XRD, which revealed that both formulations were mainly composed of calcium carbonate, calcium phosphate hydrate, zirconium oxide, and tantalum oxide. Sodium silicate hydroxide hydrate and magnesium aluminum silicate were also detected in the putty consistency, while calcium silicate sulfate and sodium aluminum silicate were detected in the paste consistency. Zamparini et al. 2018 showed a similar composition of TotalFill putty and paste; both contained calcium silicate, monobasic calcium phosphate, zirconia, tantalum peroxide, and organic components [10]. However, Moinzadeh et al. 2016 failed to detected phosphate phase in XRD [8]. The current EDX analysis showed that the regular-set paste consistency had a higher phosphorus, tantalum, and zirconium content, but less calcium than the fast-set putty. On the contrary, TotalFill paste showed less phosphorus, tantalum, and zirconium, but more calcium (atomic %), than the TotalFill putty [10].

SEM revealed that the surface of both consistencies was covered with small and rounded or small and spherulite-shaped calcium phosphate particles in the putty and paste specimens, respectively. These particles were mixed with irregular, sharp-edged particles of the calcium carbonate compound; small, spheroid, globular crystals of the calcium phosphate compound; a rod-like bundle of the calcium silicate sulfate; needle-like particles of the sulfate compounds; and radio-opacifier dots of zirconium oxide and tantalum oxide [2,14].

Apatite precipitation is an important requirement for any dental filling material. This improves the adaptation and sealing properties of filling materials [28,32]. The parameters that were used to characterize the precipitated apatite layer include the following: the intensity ratio of 965Ap/916CaWO_4_, mineral maturity [14,15,21], CO_3_/PO_4_ ratio, and crystallinity index [13,14,18]. In this study, the samples were kept in FCS for 28 days and compared with the control samples incubated in deionized water for the same period. XRD analysis detected hydroxyapatite and calcite in fast-set putty, while aragonite was exhibited in regular-set paste specimens.

The presence of ν1 PO_4_^3−^ bands at around 1030 and 965 cm^−1^ on the material-surface and material-dentin interface of both paste and putty materials after incubation in FCS indicates the formation of an apatite layer and hence the bioactivity of the material [3]. For samples that were incubated in deionized water, the ν1 PO_4_^3−^ band was originally detected at 926 cm^−1^ instead of 965 cm^−1^. The apatite crystals formed on the surface of both materials had a rosette, flower-like shape. This was confirmed by the intensity ratio of 965Ap/916CaWO_4_ and area ratio of 1030/1110 cm^−1^. However, the EndoSequence regular-set paste had a higher ratio for both parameters than the EndoSequence fast-set putty. This finding might indicate that the EndoSequence regular-set paste has a thicker apatite layer with a higher mineral maturity (transformation of non-apatitic calcium phosphate into apatite calcium phosphate) than the one formed on the surface of EndoSequence fast-set putty [16]. However, the FTIR results revealed that the EndoSequence fast-set putty had a higher crystallinity index, which indicates the perfection of apatite crystals, than the EndoSequence regular-set paste [14,30,33]. It was also revealed that the apatite layer formed on the surface of EndoSequence fast-set putty (Ca/P = 3.44) and the putty-dentin interface (Ca/P = 3.86) had a higher Ca/P ratio than that of the EndoSequence regular-set paste (Ca/P = 3.01, 1.89 respectively). Because the Ca/P ratio obtained in this study is higher than that of hydroxyapatite (Ca/P = 1.67), the precipitates seen on the surface of the studied formulations could be a combination of hydroxyapatite and calcite or aragonite [6,16,20,34]. This was also confirmed by XRD. Because the precipitate seen on the surface of the EndoSequence fast-set putty had a higher Ca/P ratio than that of the EndoSequence regular-set paste, the amount of calcite or aragonite in this precipitate could also be higher than that of the paste formulation.

It was also observed that the surface and interface of EndoSequence fast-set putty and regular-set paste had different parameters (965Ap/916CaWO4, mineral maturity, CO3/PO4, crystallinity index, and Ca/P) than those recorded after storage in FCS for 28 days. This finding suggests that the surface of both formulations was entirely covered with the precipitated apatite layer.

## 5. Conclusions

FCS incubation affects the surface morphology and chemistry of EndoSequence fast-set putty and regular-set paste root repair materials. It provides the formation of a uniform layer of flower-like crystals, which covered the entire material’s surface and the material-dentin interface in this study. Spherulites and acicular crystals were also seen at the interface. These results suggest that the precipitate is a combination of hydroxyapatite and calcite or aragonite. The EndoSequence regular-set paste had a higher mineral maturity than the EndoSequence fast-set putty. Conversely, the apatite layer seen on the EndoSequence fast-set putty had a higher crystallinity index than that of the EndoSequence regular-set paste. Consequently, both consistencies (putty and paste) of root repair material can be considered bioactive, i.e., they have an apatite formation ability when incubated in fetal calf serum, and this could be beneficial in improving the sealing ability of these materials.

## Figures and Tables

**Figure 1 materials-12-03678-f001:**
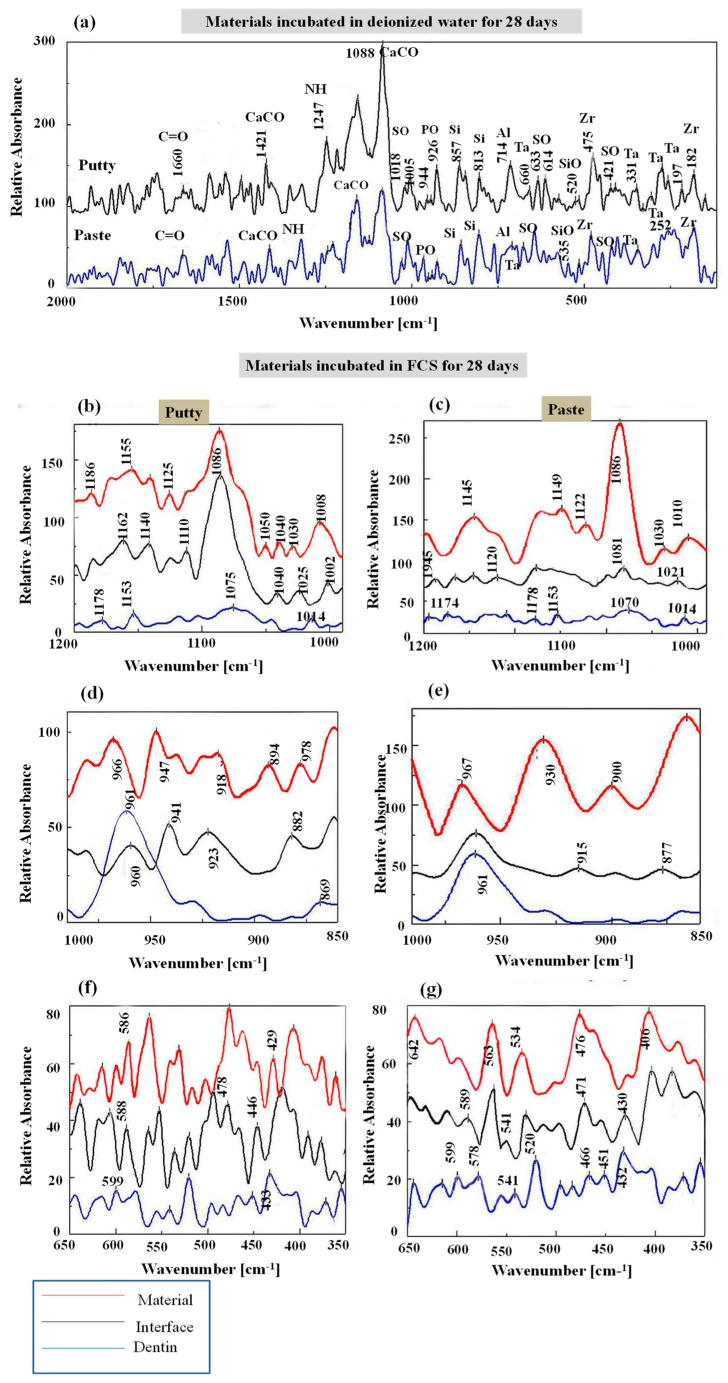
Raman spectra of EndoSequence fast-set putty versus regular-set paste kept in deionized water (**a**) or fetal calf serum (FCS) (**b**–**g**) for 28 days. Spectra include materials’ surface, materials’-dentin interface, and dentin.

**Figure 2 materials-12-03678-f002:**
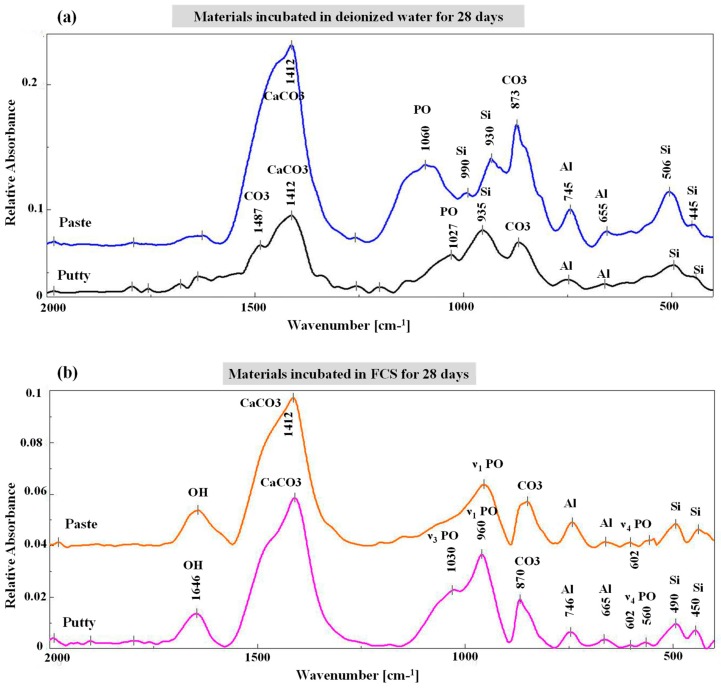
Fourier transform infrared spectroscopy (FTIR) spectra of EndoSequence fast-set putty and regular-set paste kept in deionized water (**a**) or FCS (**b**) for 28 days.

**Figure 3 materials-12-03678-f003:**
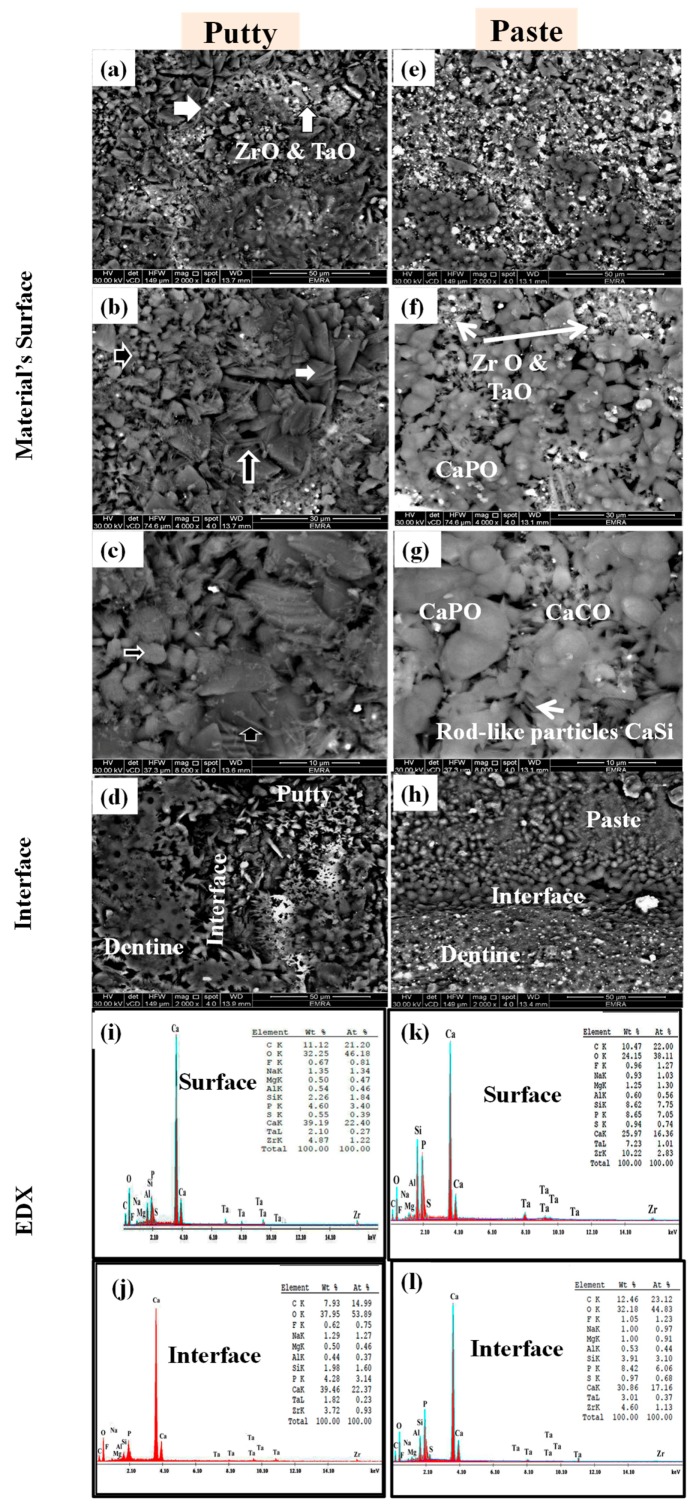
Scanning electron microscopy/energy dispersive X-ray (SEM/EDX) images of EndoSequence fast-set putty (**a**–**d**) versus regular-set paste (**e**–**h**) kept in deionized water for 28 days, as well as their dentin interface (**i**–**l**), respectively. (ZrO: zirconium oxide, TaO: tantalum oxide, CaPO: calcium phosphate, CaCO: calcium carbonate, CaSi: calcium silicate). Energy Dispersed X-ray detected bands of carbone (c), oxygen (O), fluoride (F), sodium (Na), magnesium (Mg) aluminum (Al), silicon (Si), phosphorus (P), sulfur (S), calcium(Ca), tantalum (Ta), and zirconium (Zr) on the surface and interface of both putty (**I**,**j**) and paste (**k**,**l**) specimens.

**Figure 4 materials-12-03678-f004:**
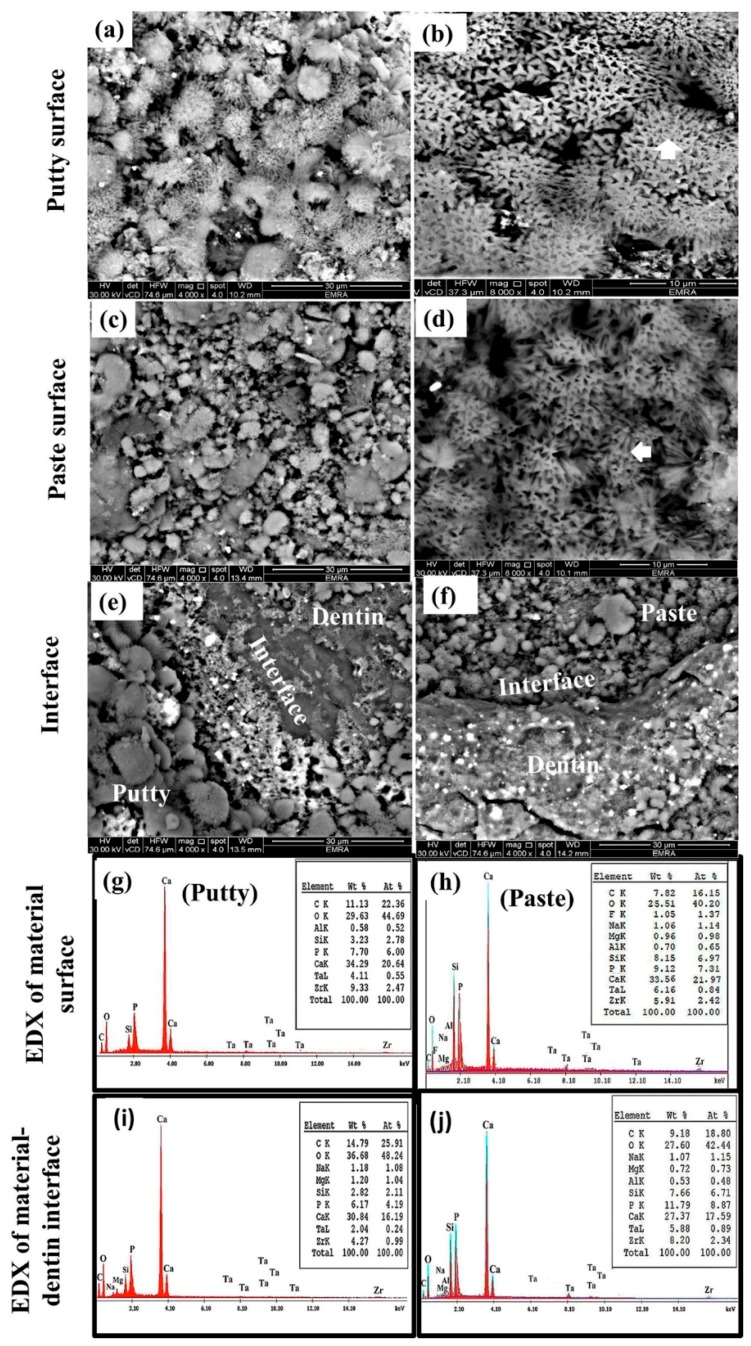
SEM images of EndoSequence fast-set putty (**a**,**b**) versus regular-set paste (**c**,**d**) kept in FCS for 28 days, as well as their interface with dentin (**e**,**f**), respectively. (**g**–**j**) represent the EDX data of the material-surface and material-dentin interface of fast-set putty and regular-set paste, respectively. (C: carbone, O: oxygen, F: fluoride, Na: sodium, Mg: magnesium, Al: aluminum, Si: silicon, P: phosphorus, S: sulfur, Ca: calcium, Ta: tantalum, and Zr: zirconium).

**Figure 5 materials-12-03678-f005:**
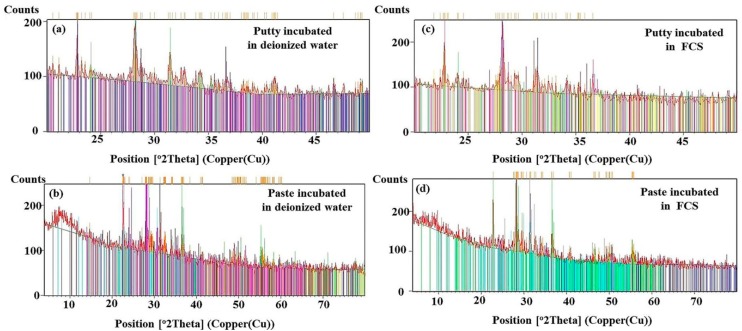
X-ray diffraction (XRD) pattern of fast-set putty and regular-set paste incubated in either deionized water (**a**,**b**) or FCS (**c**,**d**) for 28 days.

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
