# Peer review of "Morphological and Spectroscopic Study of an Apatite Layer Induced by Fast-Set Versus Regular-Set EndoSequence Root Repair Materials"

_materials, 2019, doi:10.3390/ma12223678_

Round 1

Reviewer 1 Report

The introduction is not enough to explain why these experiments were needed. In particular, why did you compare fast set type and regular set type in this experiment? You should elaborate on reason and purpose of this experiment.

Which method did you use to determine the experiment sample size? I think sample size was too small for statistical analyze.

Why did you cut the teeth longitudinally and incubated in the FCS? Clinically, the surface where the BC sealer contact with tissue fluid is limited apex area through apical foramen.

Line 114 ~ Italics are mixed in the text, causing confusion

Author Response

Comments and Suggestions for Authors

The introduction is not enough to explain why these experiments were needed. In particular, why did you compare fast set type and regular set type in this experiment? You should elaborate on reason and purpose of this experiment.

 The consistency of fast-set ( heavy consistency) is much different than regular set (flowable consistency) that affect the physical behaviour of both material e.g, their flow inside the dentinal tubules, adhesion to dentin wall, and sealing ability so it may affect their chemical behaviour and morphology as being immersed in PBS ( simulated clinical situation) . It mentioned in lines 38-42

Which method did you use to determine the experiment sample size? I think sample size was too small for statistical analyze.

We used 3 longitudinal root sections, each contains 2 holes of total 6 samples. These 6 samples for each group were analysed by FTIR, Raman spectroscopy and SEM/EDX. For each sample three readings were obtained then we have 18 reading for statistical analysis. It justified in lines 77 and 85

For XRD we use two samples just to verify the result without need for statistical analysis

Why did you cut the teeth longitudinally and incubated in the FCS? Clinically, the surface where the BC sealer contact with tissue fluid is limited apex area through apical foramen.

These two materials used as root repair materials e.g root perforation (that may be with large size) or retrograde filling so their surface were completely exposed to the tissue fluid to allow hydroxyapatite layer formation to seal the margin and improve their sealing ability

Line 114 ~ Italics are mixed in the text, causing confusion

It already corrected

Submission Date

03 October 2019

Date of this review

17 Oct 2019 06:02:49

Reviewer 2 Report

The present study evaluated the ability of root repair materials to form an surface apatite layer and investigated its structure and chemistry. The study and its results are overall of high interest to the scientific community and the experimental design is of high quality. I recommend the authors' efforts.

However, some necessary changes are to be addressed befor publication, especially regarding the introduction and the discussion.

Abstract:

l18: morphology instead of morphologic l25: "It is thicker and..." This sentence should be rewritten last senctence: this conclusion is not supported by your results, but more an interpretation (see below Discussion and Concsuions)

Introduction:

The introduction is rather short and could be improved. It starts with the word bioceramics, which is a wide field (see Donnermeyer et al., Calcium silicate based sealer -a systematic rewiew, Odontology) while the study focuses on RRM. It would be better to rewrite the beginning of this section. Also the introduction mainly deals wiht bioactivity which is not really the topic of the present study. It would be better to focus on morphology and chemistry here. Bioactivity is mor  point to be adressed in the discussion.

M&M:

l56: what kind of solution was used for moistening the gauze

l95: should be 2.6. Statistical...

Results:

l114: why the use of italic from here on???

Figure 2b: title in the figure could be more adequate -> "incubation in FCS for 28d"

Figures 3 and 4: putty and paste should be listed in diffrent columns -> exchange H and i

Discussion:

Aim of the study: Bioactivity was not assessed directly. The most part of the discussion rephrases the results without discussion them. Moreover bioactivity is a central topic of the discussion while it is not really the topic of the study.

An adequate discussion would contain:

Short presentation of the aim and the results of the study Evaluation of the hypothesis short discussion of the methods used discussion of the results: comparison to other studies, possible explanation for the differences between the groups according to the literature, possible clinical implications

Conclusion:

the conclusion should be rewritten in light of a proper discussion

Author Response

Reply to reviewer 2

The present study evaluated the ability of root repair materials to form an surface apatite layer and investigated its structure and chemistry. The study and its results are overall of high interest to the scientific community and the experimental design is of high quality. I recommend the authors' efforts.

However, some necessary changes are to be addressed befor publication, especially regarding the introduction and the discussion.

Abstract: l18: morphology instead of morphologic l25: "It is thicker and..." This sentence should be rewritten last senctence: this conclusion is not supported by your results, but more an interpretation (see below Discussion and Concsuions)

It is already corrected

Introduction:

The introduction is rather short and could be improved. It starts with the word bioceramics, which is a wide field (see Donnermeyer et al., Calcium silicate based sealer -a systematic rewiew, Odontology) while the study focuses on RRM. It would be better to rewrite the beginning of this section. Also the introduction mainly deals wiht bioactivity which is not really the topic of the present study. It would be better to focus on morphology and chemistry here. Bioactivity is mor  point to be adressed in the discussion.

Bioceramic” was changed to “Different root repair materials”

The introduction was rewrite as shown in lines 38-42

M&M: l56: what kind of solution was used for moistening the gauze

deionized water (it mention in line 60)

l95: should be 2.6. Statistical.

It done.

Results: l114: why the use of italic from here on???

It corrected

Figure 2b: title in the figure could be more adequate -> "incubation in FCS for 28d" Figures 3 and 4: putty and paste should be listed in diffrent columns -> exchange H and i

 Already corrected according to recommendation

Discussion:

Aim of the study: Bioactivity was not assessed directly. The most part of the discussion rephrases the results without discussion them. Moreover bioactivity is a central topic of the discussion while it is not really the topic of the study.

The discussion was corrected according to the reviewer recommendation

An adequate discussion would contain:

It done

Short presentation of the aim and the results of the study Evaluation of the hypothesis short discussion of the methods used discussion of the results: comparison to other studies, possible explanation for the differences between the groups according to the literature, possible clinical implications

It done

Conclusion:

the conclusion should be rewritten in light of a proper discussion

It is readjusted

Reviewer 3 Report

The authors try to evaluate the bioactivity of two root repair materials. The experiments they preset here are very basic and thus there is no novelty or significant conclusions. They should conduct more experiments in order to enrich the manuscript and provide some significant results. 

a) Many grammatical and language mistakes. Corrected examples below

L18 "This study aims to evaluate the morphological characteristics and chemistry..."

L146 "Scanning Electron Microscopy..."

b) Why the text in results section is in italics? 

c) Figure 1: The spectra for the control samples are in a different scale and thus a proper comparison cannot be made. The authors can add the respective curves for the control in each magnified figure. 

d) The compositions presented in Fig. 3 and Fig. 4 are result of an EDX map or point spectrum? If it comes from a point how many have been acquired and can the authors indicate these on the SEM images?

e) The authors should be careful regarding the Ca/P ratio because this might not correspond to the apatite layer. Depending on the thickness of the layer they examine, the underlying material is also affecting their measurement (i.e. the original Putty or paste). 

f) The XRD patterns of the milled discs are missing. No comment can be made about the analysis and discussion of this part. 

Author Response

Reply to reviewer 3

Comments and Suggestions for Authors

The authors try to evaluate the bioactivity of two root repair materials. The experiments they preset here are very basic and thus there is no novelty or significant conclusions. They should conduct more experiments in order to enrich the manuscript and provide some significant results. 

Many grammatical and language mistakes. Corrected examples below L18 "This study aims to evaluate the morphological characteristics and chemistry..." L146 "Scanning Electron Microscopy..." Why the text in results section is in italics? 

it corrected

c)Figure 1: The spectra for the control samples are in a different scale and thus a proper comparison cannot be made. The authors can add the respective curves for the control in each magnified figure.

The figure 1 was adjusted as recommended with similar scale. The control spectra (samples incubated in water) is already mention (Figure 1 a) 

d)The compositions presented in Fig. 3 and Fig. 4 are result of an EDX map or point spectrum? If it comes from a point how many have been acquired and can the authors indicate these on the SEM images?

The composition of figures 3 and 4 are related to EDX map. It was taken from three different areas across the surface.

e)The authors should be careful regarding the Ca/P ratio because this might not correspond to the apatite layer. Depending on the thickness of the layer they examine, the underlying material is also affecting their measurement (i.e. the original Putty or paste).

It was corrected in line 242.

We agree with reviewer comment regarding Ca/P ratio, but looking at Ca/P ratio for both control and experimental samples there is significant difference between both of them. Putty showed higher Ca/P ratio that might indicate a layer of sufficient thickness is precipitated on putty surface. The changes of Ca/P ratio may be related to increase of phosphate constituents after apatite precipitation.

f)The XRD patterns of the milled discs are missing. No comment can be made about the analysis and discussion of this part.

The XRD pattern was added (Figure 5). The analysis of material incubated in water was discussed in line 223-229 and 245 for material incubated in FCS.

Round 2

Reviewer 1 Report

As you written, these materials usually used for retrograde filling and perforation repair. So, I think horizontal section is more clinical relevance experiment design, not longitudinal.

Author Response

Replay to reviewer 1

As you written, these materials usually used for retrograde filling and perforation repair. So, I think horizontal section is more clinical relevance experiment design, not longitudinal.

The authors agree with the reviewer to perform horizontal section, if the study is subjected to retrograde situation or used root canal sealers. But this study was focused to use the ERRM as repair material the root perforation defect. The perforation most commonly occurs at lateral root surface or furcal perforation, and this articles focused on material used as repair for root perforation not as retrograde or root canal sealers. When the root perforation occurs on the lateral root surface, longitudinal root surface is more clinically relevant When the perforation occurs at furcal area, dentin block is more clinically relevant. The study design was update with reference (Line 77 track change)

Introduction section was update to describe the background of study and explain the aim of this study “as shown in track change version” Regarding to FCS, The serum is liquid portion of the blood that allow apatite formation. In our study, FCS used according to previous study Ishikawa et al 1994 & Tingey et al 2008.(Line 64 track change)

Ishikawa et al 1994: Behavior of a calcium phosphate cement in simulated blood plasma in vitro; Dent Mater 10:26-32,1994;

Tingey MC, Bush P, Levine MS. Analysis of mineral trioxide aggregate surface when set in the presence of fetal bovine serum. Journal of endodontics. 2008;34(1):45-9.

Regarding the sample sized, Sample size was selected according to zampirini et al 2018. This reference was included in M & M section. (Line 77 track change)

Zamparini F, Siboni F, Prati C, Taddei P, Gandolfi MG. Properties of calcium silicate-monobasic calcium phosphate materials for endodontics containing tantalum pentoxide and zirconium oxide. Clinl Oral Investig. 2018:1-13.

 Spelling mistaks were carried out and manuscript has been edited

Reviewer 2 Report

All necessary changes were made in the manuscript.

Author Response

Replay to reviewer 2

Spelling mistaks and english editing were carried out and manuscript has been editing

Reviewer 3 Report

The authors tried to comply with almost all of the comments. As i mentioned to my initial review the results are lacking in novelty (no additional experiments have been conducted). However the manuscript now is improved.